# Physical Aging Behavior of a Glassy Polyether

**DOI:** 10.3390/polym13060954

**Published:** 2021-03-20

**Authors:** Xavier Monnier, Sara Marina, Xabier Lopez de Pariza, Haritz Sardón, Jaime Martin, Daniele Cangialosi

**Affiliations:** 1Donostia International Physics Center, Paseo Manuel de Lardizabal 4, 20018 San Sebastián, Spain; xavier.monnier.90@gmail.com; 2POLYMAT, University of the Basque Country UPV/EHU, Av. de Tolosa 72, 20018 San Sebastián, Spain; sara.marina@polymat.eu (S.M.); xlopezdepariza001@ikasle.ehu.eus (X.L.d.P.); haritz.sardon@ehu.es (H.S.); jaime.martin@polymat.eu (J.M.); 3Ikerbasque, Basque Foundation for Science, 48013 Bilbao, Spain; 4Centro de Investigacións Tecnolóxicas, Universidade da Coruña, Campus de Esteiro s/n, 15403 Ferrol, Spain; 5Centro de Fisica de Materiales (CSIC-UPV/EHU) Paseo Manuel de Lardizabal 5, 20018 Sebastián, Spain

**Keywords:** calorimetry, physical aging, glasses

## Abstract

The present work aims to provide insights on recent findings indicating the presence of multiple equilibration mechanisms in physical aging of glasses. To this aim, we have investigated a glass forming polyether, poly(1-4 cyclohexane di-methanol) (PCDM), by following the evolution of the enthalpic state during physical aging by fast scanning calorimetry (FSC). The main results of our study indicate that physical aging persists at temperatures way below the glass transition temperature and, in a narrow temperature range, is characterized by a two steps evolution of the enthalpic state. Altogether, our results indicate that the simple old-standing view of physical aging as triggered by the α relaxation does not hold true when aging is carried out deep in the glassy state.

## 1. Introduction

A glass can be formed by several routes, all of them sharing the prerequisite of allowing circumventing crystallization. Among them, that based on cooling through the melting, $T_{M}$, and the glass transition, $T_{g}$, temperatures is by far the most common [1]. The kinetic nature of the glass transition, well exemplified by the cooling rate dependence of $T_{g}$ [2,3], implies that glasses are thermodynamically in non-equilibrium. The slow evolution of the thermodynamic state toward the metastable equilibrium state represented by the supercooled liquid is generally addressed as structural recovery [4] or physical aging [5,6,7,8]. This phenomenon induces a general time-dependent modification of the glass properties and, therefore, is of utmost importance from both fundamental and technological viewpoints. Indeed, volume shrinkage could be detrimental for the glass lifetime [9]; therefore, aging must be by glass manufacturers to avoid any undesired alteration of properties over the course of time. Notable examples in this sense are those on the effect of aging on gas transport properties [10,11,12,13] also in relation to mechanical properties [14]. The relation between gas transport properties and the thermodynamic state of the glass has also been discussed [15]. Furthermore, knowledge of physical aging can provide insights of utmost importance on dynamics and thermodynamics of glasses sitting at the bottom of the energy landscape [16,17].

The conventional belief, based on the archetypal volume recovery experiments of Kovacs [4] and later by a wealth of experiments [6,7], is that recovery of equilibrium takes place with a monotonous sigmoidal-shaped evolution of the glass thermodynamic state and that such evolution is triggered exclusively by the main relaxation process, generally addressed as ”α relaxation”, exhibiting super-Arrhneius behavior and diverging not far below $T_{g}$. The common features of studies showing this behavior is that physical aging is carried out either in proximity of $T_{g}$ or, if aging temperatures considerably smaller than $T_{g}$ are considered, for aging times not long enough to allow attaining the final equilibrium.

Recently, experiments by differential scanning calorimetry (DSC)—where the evolution of the enthalpy of the glass is monitored—showed that, if physical aging is conducted considerably below $T_{g}$ and for aging times as long as about one year, two steps in the approach to equilibrium of the enthalpy, each characterized by the attainment of a plateau, are observed for different polymers [18]. This event was shown independently in other glasses, including chalcogenides [19], a small molecule [20], metallic glasses [21,22], and polysulfone [23]; and variously modeled according to different approaches [24,25,26]. Furthermore, if aging is conducted far below $T_{g}$, prolonged aging results in the attainment of partial recovery of equilibrium even though a plateau in the enthalpy is achieved [27]. The presence of multiple steps indicates that there exist different mechanisms of equilibrium recovery, whose existence is evidenced by the thermal response of glasses aged well below $T_{g}$. Specifically, in these cases, specific heat scans show the presence of an excess endotherm of the aged sample with respect to the unaged one. This outcome appears to be general, as it was found in glasses of different nature [28,29], including polymers [27,30,31], metallic glasses [29,32], and a plastic crystal [28]. Importantly, this behavior is magnified in polymer glasses exhibiting large free interfacial area [33,34], as a result of the acceleration of physical aging in these systems [35,36,37,38,39], which amplifies the separation among different mechanisms of equilibrium recovery [36]. Despite the variety of experiments showing these features, results showing several steps in isothermal conditions are relatively scarce. The main reason is likely that, to observe this behavior, a wide interval of aging times is required. The lower aging time bound attainable by standard calorimetric techniques is typically of the order of minutes. Hence, to cover an aging time interval of several decades (for instance, 5), time scales of months to years are required [18,19,27,29].

In the present work, we employ the capabilities of fast scanning calorimetry (FSC) [40,41] permitting to access heating/cooling rates of the order of several thousands kelvin per second to study the physical aging behavior in an amorphous polyether. By reducing the time scale of the experiments over orders of magnitude with respect to standard calorimetry, FSC allows accessing sub-second evolution of physical aging. Furthermore, samples heated at high rates are less amenable to chemical degradation, since the time spent at high temperatures is very short. This allows investigating phenomena, including glass transition [42,43], melting [44], and polymer adsorption [45], otherwise impossible to study by standard calorimetry. The choice of an amorphous polyether rests on his chemical difference with previously employed polymers [18], which were either vinyl polymers or presenting an aromatic ring in the backbone. We find that, similarly to other amorphous polymers with substantially different molecular structure [18,23], physical aging exhibits two mechanisms of equilibration, which, in the case of isothermal experiments, is indicated by the presence of two decays towards equilibrium.

## 2. Materials and Methods

### 2.1. Materials

Poly(1,4-cyclohexanedimethanol) (PCDM), whose chemical structure is reported in Figure 1, was synthesized by polycondensation of 1,4-cyclohexanedimethanol (CHDM) using the previously reported non-eutectic acid-base organocatalysts based on MSA and TBD (3:1) [46,47]. It is worth noting that the employed CHDM contains a cis:trans isomer mixture equal to 70:30, that strongly inhibits the crystallization of the material. In a typical reaction, CHDM was polymerized in the presence of the catalyst (7.5 and 2.5 mol % of MSA and TBD, respectively, with respect to the monomer) for 72 h with a gradual increase of temperature from 130 to 200 ${}^{\circ }$C under vacuum. The resulting product was purified by precipitation in cold MeOH from a CHCl3 solution. This procedure was repeated three times to yield the pure homopolymer that was characterized by 1H NMR (300 MHz, CDCl3 δ): = 3.58–3.49 (1H), 3.31–3.10 (16H), 1.84 (16H), 1.5–1.3 (15H) 0.95 (13H). 13C NMR (300 MHz, CDCl3 δ): = 77.9, 39.7, 30.7. GPC. (THF, 25 ${}^{\circ }$C) Mn: 10 kDa, Đ = 1.8.

### 2.2. Methods

The kinetics of physical aging was studied following the evolution of the enthalpy by fast scanning calorimetry. To this aim, the Flash DSC-1 by Mettler-Toledo (Nänikon, Switzerland), based on chip calorimetry technology, was employed, allowing heating/cooling rates as large as more than 1000 K/s. Thanks to the fast temperature stabilization, such high rate allows investigating aging from time scales in the sub-seconds range, i.e., way below the time scales accessible by standard calorimetry. Measurements were carried out under nitrogen gas flux with rate of 20 mL min${}^{-1}$ and the temperature was controlled by a two stage intracooler, allowing for temperature control between −90 and 500 ${}^{\circ }$. PCDM samples were directly placed on the active area of the chip. The mass was, in all cases, about 100 ng. For all samples, within the experimental uncertainty, we obtained identical results, indicating that size effects, previously observed for samples smaller than 100 ng in another polymer [48], were of no relevance for the relatively large samples of our study.

All measurements were set to begin at 50 ${}^{\circ }$C, i.e., well above PCDM $T_{g}$ (−6 ${}^{\circ }$C on cooling at 1000 K/s). Samples were subsequently cooled with 1000 K/s to the selected aging temperature, $t_{ag}$, ranging from just below to far below $T_{g}$, i.e., from −10 ${}^{\circ }$C to −70 ${}^{\circ }$C with intervals of 5 ${}^{\circ }$C. Once stabilized at $T_{ag}$, samples were aged for different aging times, $t_{ag}$, ranging from the 0.1 to ∼10${}^{5}$ s, which implies that our determination of the kinetics of aging covered about 6 decades in time. After aging, samples were immediately cooled to −90 ${}^{\circ }$C and re-heated to 50 ${}^{\circ }$C. In both cases, the heating rate was 1000 K/s. Before and after the aging cycle, a reference scan was obtained by heating at 1000 K/s a sample previously directly cooled from 50 to −90 ${}^{\circ }$C. A scheme of the employed thermal protocol is shown in Figure 2.

In non-equilibrium glasses, a common way to quantify the thermodynamic state attained after a given thermal protocol relies on the concept of fictive temperature, $T_{f}$, defined as the temperature corresponding to the intercept of the extrapolated glass and supercooled equilibrium enthalpic lines [49]. Operationally, considering that the first derivative at constant pressure of the enthalpy equals the specific heat, $C_{p}$, $T_{f}$ is obtained via the method of the matching areas proposed by Moynihan and co-workers [50]:(1)$$\int_{T_{f}}^{T>>T_{g}}\left(C_{pm}-C_{pg}\right)dT=\int_{T<<T_{g}}^{T>>T_{g}}\left(C_{p}-C_{pg}\right)dT,$$
where $C_{pm}$ and $C_{pg}$ are the supercooled melt and glass specific heats, respectively. FSC, similarly to standard calorimetric techniques, delivers the heat flow rate, *W*, i.e., the specific heat per sample mass, *m*, and applied heating/cooling rate, β: $W=C_{p}m\beta$. Hence, Equation (Equation 1) can be rewritten in terms of the heat flow rate:(2)$$\int_{T_{f}}^{T>>T_{g}}\left(W_{m}-W_{g}\right)dT=\int_{T<<T_{g}}^{T>>T_{g}}\left(W-W_{g}\right)dT,$$
where $W_{m}$ and $W_{g}$ are the heat flow rates of the supercooled melt and the glass, respectively. From a graphical viewpoint, the way $T_{f}$ is obtained from specific heat scans is described in the seminal work of Moynihan and co-workers [50] and redrawn in a recent review [2].

## 3. Results

The heat flow rate as a function of temperature at 1000 K/s for different aging times and at some selected aging temperatures is shown in Figure 3. In all cases, aging induces the development of an endothermic overshoot whose size increases with aging time. Aging at temperatures not too far from $T_{g}$ results in the standard behavior with a narrow overshoot in proximity of the step of the glass transition. When the aging temperature is decreased, the endothermic overshoot progressively shifts to lower temperature. At −45 ${}^{\circ }$C, for short aging times, the endothermic overshoot is well separated from the step at the glass transition, whereas, for longer aging times, it progressively overlaps with such step, similarly to aging at higher temperature. Aging at even lower temperatures shows the endothermic overshoot resulting from physical aging at an even lower temperature range, well separated from the step at the glass transition. In these cases, no signature of aging is present in proximity of $T_{g}$.

A way to obtain insights on the physical aging behavior relies on the isochronal representation, where the thermodynamic state in terms of $T_{f}$ is shown as a function of the aging temperature at a fixed aging time. This is shown in Figure 4 for two aging times, where the difference of the polymer $T_{g}$ to the $T_{f}$ obtained after given aging conditions is shown. This difference is zero in unaged samples. First inspection of the figure shows the typical signature of physical aging, i.e., a maximum in the reduction of $T_{f}$, in line with previous reports [36,37,51,52]. This behavior is explained considering that the physical aging depends on two counteracting factors: the molecular mobility triggering the evolution of the thermodynamic state and the distance of the glass from equilibrium, i.e., the thermodynamic driving force. Close to $T_{g}$, the former factor is large, being the glass at relatively high temperature. Hence, the glass will rapidly evolve towards equilibrium. However, the distance from equilibrium will be small; therefore, only limited reduction of $T_{f}$ are allowed. Decreasing the aging temperature entails an increase of the thermodynamic driving force, which implies an increase of $T_{g}-T_{f}$ to a maximum. For aging times of 10 and 1000 s (see Figure 4), this maximum is located at ∼−15÷ −25 ${}^{\circ }$C. The presence of a temperature maximum in the amount of recovered enthalpy implies that there exists an optimum aging temperature to maximize equilibrium recovery at fixed aging time [53]. At lower aging temperatures, the slowing down of molecular mobility becomes dominant; therefore, the reduction in $T_{f}$ over limited aging time scales progressively decreases. Importantly, this reduction takes place smoothly over a temperature range of dozens of degrees. In the case of samples aged for 1000 s, a decrease in $T_{f}$ persists even at $T_{ag}=T_{g}-∼100{\;}^{\circ }$C. As a result, aging way far from $T_{g}$ is not suppressed as one would expect if only the α relaxation, whose time scale diverges below $T_{g}$, was considered.

A different viewpoint of the aging behavior can be drawn from the isothermal representation, which consists of monitoring the time evolution of $T_{f}$ at different temperatures. This is shown in Figure 5 for different selected aging temperatures. Close to $T_{g}$, aging exhibits the standard behavior consisting of a monotonous decay toward equilibrium, which is marked by the condition $T_{f}=T_{ag}$. However, at lower temperatures, specifically −22 and −26 ${}^{\circ }$C in Figure 5, aging takes place with an intermediate plateau with $T_{f}>T_{ag}$. Furthermore, in the case of $T_{ag}=$ −22 ${}^{\circ }$C, beside the first step, the complete recovery of equilibrium with a second decay can be observed.

To obtain a typical time scale for physical aging, a model-independent route consists of considering the aging time needed to attain each plateau in $T_{f}$. This is shown by arrows in Figure 5. In the conditions shown in this figure, only one time scale is obtained at the highest investigated aging temperatures. At −22 ${}^{\circ }$C, two time scales can be detected, whereas, at lower temperatures, either only the time scale of the fast process of equilibrium recovery or no time scale at all can be detected. An overview of the kinetics of equilibrium recovery, in terms of time scale to reach equilibrium, $\tau_{eq}$, is shown in Figure 6. At high temperatures, i.e., close to $T_{g}$, only one time scale, associated to the monotonous decay to equilibrium, can be observed. Lower aging temperatures entail a split into two time scales: (i) the slow one, associated to the α relaxation, with $\tau_{eq}$ increasing in a super-Arrhenius fashion with decreasing temperature, thereby becoming to large to be observed not far below $T_{g}$; and (ii) a fast time scale exhibiting milder temperature dependence and seemingly decreasing activation energy with decreasing temperature.

## 4. Discussion

The paradigmatic view of physical aging describing it as triggered by the α relaxation, in which typical time scale increases to unfeasible values not too far below $T_{g}$ as a result of its super-Arrhenius temperature dependence, has conditioned experimental activity over long time [6]. Such view is generally accurate enough to describe physical aging in proximity of $T_{g}$. However, a number of experimental observations dating back several decades ago already pointed towards the incompatibility of this view with the persistence of physical aging when carried out way below $T_{g}$ [5,54,55,56]. Our isochoric experiments (see Figure 4) are in line with those previous reports, as they show that, even at $T_{g}-∼90{\;}^{\circ }$C, mild conditions of aging, in the case of Figure 4, $t_{ag}=10^{3}$ s, result in a significant drop of $T_{f}$. In calorimetry, this effect is generally associated with a low temperature endothermic overshoot [27,30,57,58], in ways analogous to what it is shown in Figure 3. The latter underlines a devitrification step taking place at temperatures at which the α relaxation, increasing to extremely large time scales at temperatures not far below $T_{g}$, is of no relevance. In contrast, as shown in the kinetic plot of Figure 6, a finite time scale associated to the fast mechanism of equilibrium recovery persists even far below $T_{g}$, thereby triggering early devitrification of glasses aged via such mechanism.

While studies showing the presence of aging well above $T_{g}$ and/or its associated low temperature endothermic overshoot are numerous, those where the two step decays of enthalpy or equivalently $T_{f}$ in bulk glasses are relatively scarce [18,21,23]. The reason becomes immediately clear observing the kinetic plot of Figure 6. If an aging temperature close to $T_{g}$ is chosen, only one equilibration step associated to the α relaxation is present. Aging at lower temperatures allows splitting the fast mechanism of equilibrium recovery from that of the α relaxation. However, the possibility of identifying the two steps of equilibration rapidly fades away on further cooling as a result of the rapidity of increase of equilibration time associated to the α relaxation. With these premises in mind, in our study we show, for the first time by FSC on a bulk polymeric glass former, that the presence of multiple equilibrium recovery steps can be identified in a tiny aging temperature range, in this case between −22 and −26 ${}^{\circ }$C. Lower aging temperatures entail a slowing down to unfeasible time scales of the mechanism associated to the α relaxation. However, Figure 5 shows that aging persists on time scales shorter than 10${}^{5}$ s at temperatures well below $T_{g}$. A wealth of studies on a wide variety of polymeric glass formers showed that longer aging times result in the attainment of a plateau with $T_{f}>T_{ag}$, i.e., with partial recovery of equilibrium [27,59,60,61,62,63,64]. In one of these studies [63], experiments conducted within the so-called “asymmetry of approach” thermal protocol indicated that such plateau corresponds to a relative minimum in the free energy, whose access is triggered by the fast mechanism of equilibration.

Importantly, the presence of a fast mechanism of equilibrium recovery is found in PCDM, i.e., a polymer with relatively simple chemical structure (see Figure 1). This result suggests that the origin of complex aging behavior must be based on a framework disregarding the chemical details of the glass. Indeed, the presence of a fast mechanism of equilibrium recovery has been encountered on the base of a variety of observations in a wide range of system, including bulk metallic glasses [21,22,29,32,65,66,67], phase change materials [68], a plastic crystal [28], chalcogenide glasses [19], and a low molecular weight glass former [20]. Although glass-former specific approaches are of interests [25], these results calls for a universal description. Among the different approaches aiming to describe the presence of different mechanisms of equilibration, some rely on pure phenomenological description [26,69]. On more theoretical grounds, the shear transformation zone theory [70], the concept of marginal glass [71], and the random first order transition (RFOT) [24] of the glass transition provide a picture in which there exist multiple states in the glass and/or the presence of zones with enhanced mobility. The possibility to incorporate the presence of multiple mechanisms of equilibration within the Adam-Gibbs framework [72], where different cooperative length scale would be associated to each mechanism, could also deserve attention [73]. In this sense, experimental advances in the search of the molecular mechanism responsible for aging way below $T_{g}$ are warranted.

Finally, it is worthy pointing out that the presence of multiple mechanisms of equilibration could be of relevance to describe a wide range of non-equilibrium phenomena in amorphous polymers [74]. This has been, for instance, put forward in the shape recovery kinetics of shape memory polymers (SMP) [75]. Other examples where the kinetic of equilibration may be triggered by either the α relaxation or another mechanisms with milder activation energy are polymers adsorbed on an substrate [45,76] and polymer films dewetting [77].

## 5. Conclusions

The vast majority of physical aging experiments are conducted in proximity of $T_{g}$, where approach to equilibrium is triggered by the primary α relaxation. Our study takes inspiration by recent findings showing the existence of other mechanisms of equilibrium recovery deep in the glassy. To this aim, we have studied the physical aging behavior of a glassy polyether, PCDM, by fast scanning calorimetry, which allow characterizing the time dependent evolution of the thermodynamic states in terms of $T_{f}$. Our results indicates that physical aging persists at temperatures way below $T_{g}$ and, at these low temperatures, manifests with a low temperature endothermic overshoot. Furthermore, we show that, in a narrow temperature range, somewhat below $T_{g}$, two steps in the approach to equilibrium can be detected. Altogether, our results provide compelling evidence for the presence, beyond the α relaxation, of a second mechanism of equilibrium recovery. The nature of this mechanism can be described invoking different theoretical approaches, while, at the same time, efforts in the identification of its molecular nature are warranted.

## Figures and Tables

**Figure 1 polymers-13-00954-f001:**
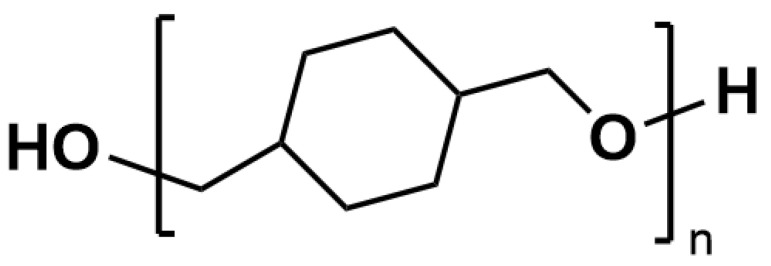
Chemical structure of PCDM.

**Figure 2 polymers-13-00954-f002:**
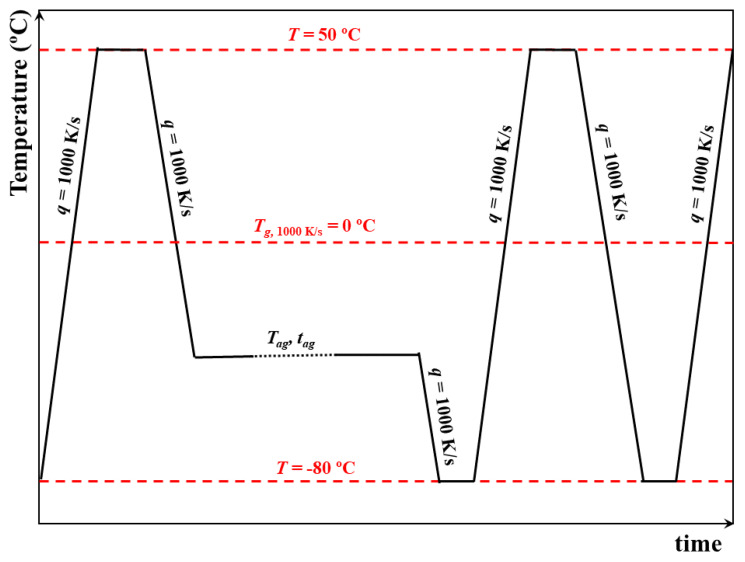
Thermal protocol employed to study physical aging in PCDM.

**Figure 3 polymers-13-00954-f003:**
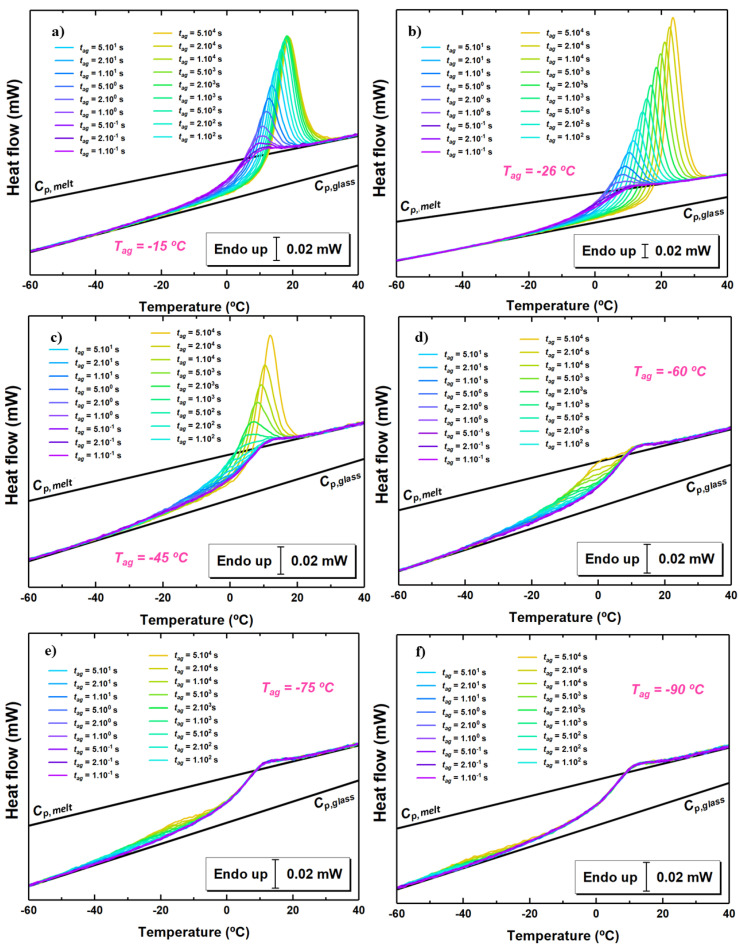
Heat flow rate scans at the indicated times and the following aging temperatures: (**a**) −15 ${}^{\circ }$C; (**b**) −26 ${}^{\circ }$C; (**c**) −45 ${}^{\circ }$C; (**d**) −60 ${}^{\circ }$C; (**e**) −75 ${}^{\circ }$C; (**f**) −90 ${}^{\circ }$C.

**Figure 4 polymers-13-00954-f004:**
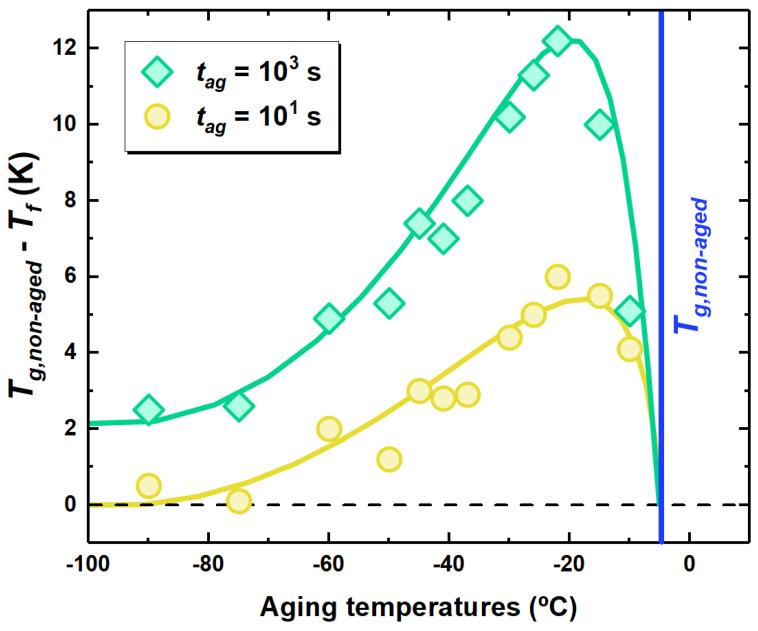
Evolution of $T_{f}$, taken as the distance from $T_{g}$, with the aging temperature for PCDM aged at the indicated aging times.

**Figure 5 polymers-13-00954-f005:**
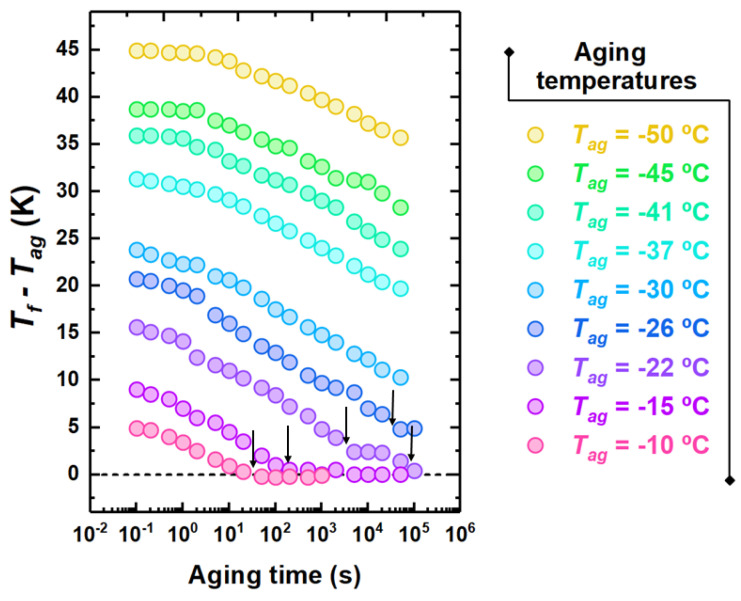
Evolution of $T_{f}$ taken as the distance from $T_{g}$ with the aging time for PCDM aged at the indicated aging temperatures. The arrows indicate the time scale to reach the plateau.

**Figure 6 polymers-13-00954-f006:**
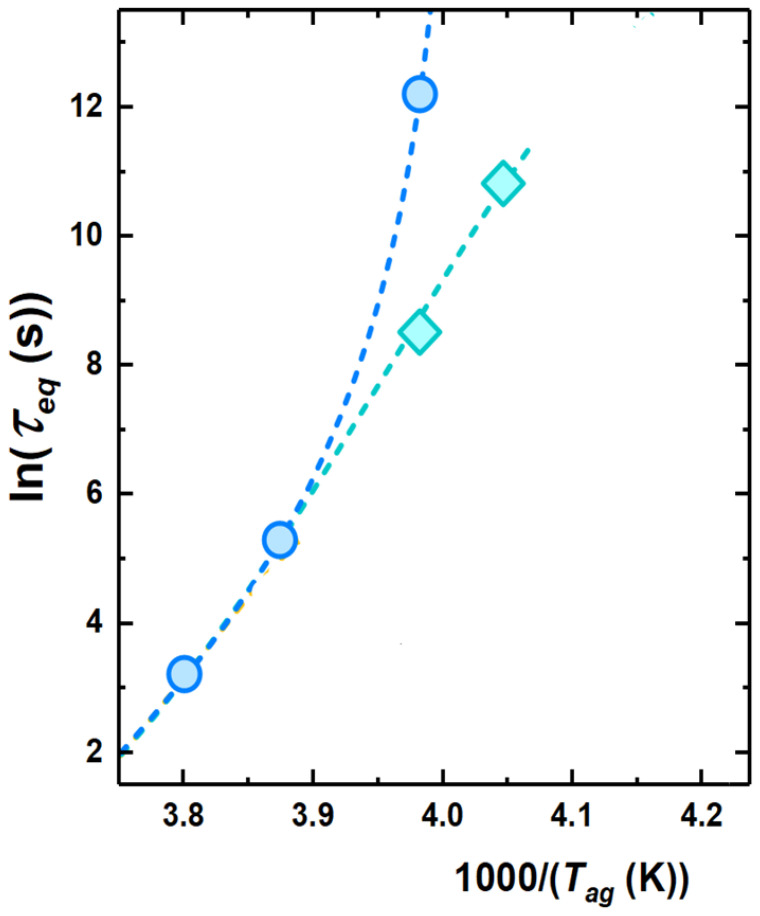
Activation plot showing the temperature evolution of the two time scales to reach equilibrium.

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
