# Peer review of "Physical Aging Behavior of a Glassy Polyether"

_polymers, 2021, doi:10.3390/polym13060954_

Round 1

Reviewer 1 Report

First of all I must say, it is a well-organised presentation of some results on physical ageing of a particular polymer. In this point, the manuscript can be accepted. However, I have few observations that need to be addressed first.

  1. I was reading the introduction carefully. However, I do not find a single sentence why we need to study the physical ageing phenomenon. Moreover, why we need to study the physical ageing of polyether is also absent in the introduction.
  2. It is not clear why the particular cooling rate (1000k /s, Line 76) is chosen
  3. Please be consistent in nomenclature (T_a and t_a are the same)
  4. I would prefer to see the same font size and shape of Graphs in 3 and 4
  5. Please note that modeling of different ageing phenomena is also an active area of research. So few literature  from that community will surely widen the reach  of the current work. https://doi.org/10.1016/j.jnoncrysol.2018.10.021
  6. https://doi.org/10.1016/bs.aams.2015.10.003

Author Response

Our response to this Reviewer is contained in the uploaded pdf file.

Reviewer 2 Report

This manuscript describes a systematic study on the effect of aging time and temperature and of the scanning rate on the glass transition temperature of a glassy poly(1-4 cyclohexane di-methanol) by fast scanning DSC. The work is carried out well, very systematically, and the discussion is sound. Therefore I can recommend this work for publication. However there are some points to be solved before the manuscript can be accepted.

In general, neither the introduction of the fast scanning technique, nor the discussion of materails properties is very strong.

Some more introduction on the limits and advantages of fast scanning DSC could be useful. For instance, the advantage of extremely fast analysis which allows the analysis of glass transition temperatures even for materials with a Tg above the degradation temperature [H. Yin, Y.Z. Chua, B. Yang, C. Schick, W.J. Harrison, P.M. Budd, M. Böhning, A. Schönhals, First Clear-Cut Experimental Evidence of a Glass Transition in a Polymer with Intrinsic Microporosity: PIM-1, J. Phys. Chem. Lett. 9 (2018) 2003–2008. doi:10.1021/acs.jpclett.8b00422.]

Furthermore, some more discussion on other polymer properties and the correlation with aging would increase the insight of the readers. For instance, there is a vast amount of literature related to the correlation between physical aging and gas transport, one of the most sensitive probes for the analysis of physical aging in polymeric membranes and physical aging, which discusses the physical aging in terms of changes in the excess free volume of the polymer. Thornton et al, and Rowe et al report a mechanism at two different time scales [A.W. Thornton, A.J. Hill, Vacancy Diffusion with Time-Dependent Length Scale: An Insightful New Model for Physical Aging in Polymers, Ind. Eng. Chem. Res. 49 (2010) 12119–12124. doi:10.1021/ie100696t; B.W. Rowe, S.J. Pas, A.J. Hill, R. Suzuki, B.D. Freeman, D.R. Paul, A variable energy positron annihilation lifetime spectroscopy study of physical aging in thin glassy polymer films, Polymer (Guildf). 50 (2009) 6149–6156. doi:10.1016/j.polymer.2009.10.045.]. Can the present method help in the interpretation of these works? Furthermore, Longo et al. also correlate aging to the young’s modulus, which drastically increases upon aging. The article would improve if the authors would discuss somewhat more the physical background and the polymer properties, rather than only the thermal properties.

Specific points

  1. The properties of thin films may be strongly affected by interfacial phenomena (free surface or adhesion to a support) and these may be significantly different from the bulk polymer properties. Given the extremely small size of samples in fast scanning DSC, can we still speak about bulk polymer properties?

  1. On p. 3 the authors report aging times as short as 0.1 s. Is this feasible? Is the cooling rate really that fast?

  1. Page 3-4, Eqs. 1 and 2. It would be helpful to illustrate this procedure graphically.

  1. Eq. 2: Wm is defined as the heat flow rate of the melt. Would it not be more correct to speak about the ‘rubber’ instead of the ‘melt’, since the Tg is the glass transition temperature and not the melting temperature.

  1. Similarly, in Fig. 3 the phase above Tg is defined as the liquid phase. Rheological measurements should be needed to establish whether the viscous (liquid) or elastic (rubber) behaviour of the material dominates just above Tg. I do not think that the latter is very likely. Are these data available? Is the molar mass of the polymer known? If this is high enough, then liquid behaviour can be excluded.

Minor issues

  1. Abstract: poly(1-4 cycloexane di-methanol): ‘exane’ must be ‘hexane’

  1. Both T_a and T_ag are used for the aging temperature. Please use only one symbol

  1. The line of Tau_eq2 in Fig 6 should not be extrapolated that far beyond the experimental data, since there is no evidence for the curve shape in the region above 1/T = 4.05

  1. Use decimal points instead of decimal commas in Fig 6

Author Response

(The authors gave the same response as above.)

Reviewer 3 Report

The study experimentally examines the aging behavior of a glassy polyether. This is an important field. The results show some interesting contribution to understand the multiple equilibration mechanisms in physical aging of glasses. Very well written paper.  Easy to follow and flows very well.  This reviewer can recommend for publication in the present form.

Author Response

(The authors gave the same response as above.)

Reviewer 4 Report

The present study reports on physical aging behavior of a glassy polyether, and two working principles and equilibration mechanisms in physical aging have been experimentally detected and discussed. After carefully reading it, I would like to suggest to consider the following points.

1. “The kinetic nature of the glass transition, well exemplified by the cooling rate dependence of Tg [2,3],” I can not agree with this explanation. As for Tg, it is determined by the heating rate, while the T0 is determined by the cooling rate. As presented in ref. [21], the effect of prior history on enthalpy relaxation in glassy polymers has been presented that.

2. The time scale in figure 2 is suggested to mark for it.

3. The Tg values in figure 3 are suggested to plot in a new figure for comparison.

4. For the discussion on the equilibration mechanisms in physical aging, please consider the reference, Adam-Gibbs Formulation of Enthalpy Relaxation Near the Glass Transition (https://nvlpubs.nist.gov/nistpubs/jres/102/2/j22hod.pdf). Generally, there is a strong cooperative interaction for the glass transition due to the discreteness of various recovery of segments and macromolecular chains. Meanwhile, please consider the effect of free volume on the equilibration mechanisms in physical aging, generally, it is popular to catch up with the discussion of this study by means of “free-volume” theory. Whether the Arrhenius rule or super-Arrhenius rule, it is experimentally determined by the free volume, for the physical aging. The following reference is helpful for it, (1) Haibao Lu and Shanyi Du. A phenomenological thermodynamic model for the chemo-responsive shape memory effect in polymers based on Flory-Huggins solution theory. Polymer Chemistry. 2014, 5(4), 1155-1162.

5. I suggest to cover the effect of component on the physical aging and interval of aging times. According to the Gordon-Taylor rule, the Tg is originated from the interaction of the components in macromolecular chain. And then the physical aging is therefore critically determined by the components, which determines the separation, interface and enthalpy relaxation, i.e., (1) Haibao Lu and Wei Min Huang. On the origin of the Vogel-Fulcher-Tammann law in the thermo-responsive shape memory effect of amorphous polymers. Smart Materials & Structures. 2013, 22(10): 105021.

In all, it is a good job. For the Tg and physical aging matters, they are so difficult to promote the research in the last sixty years. However, this study presents a new design and explanation on it, I think I would like to recommend it, and it will attract many readers.

Author Response

(The authors gave the same response as above.)
